# Effect of TiO_2_ Nanoparticle on Bioaccumulation of ndl-PCBs in Mediterranean Mussels (*Mitilus galloprovincialis*)

**DOI:** 10.3390/ani13071208

**Published:** 2023-03-30

**Authors:** Federica Gallocchio, Alessandra Moressa, Francesco Pascoli, Alessia Vetri, Anna Toffan, Tobia Pretto, Giuseppe Arcangeli, Roberto Angeletti, Antonia Ricci

**Affiliations:** 1Laboratorio di Chimica, Laboratorio Nazionale di Riferimento per i Nanomateriali Negli Alimenti, Istituto Zooprofilattico Sperimentale delle Venezie, 35020 Padova, Italy; 2Laboratorio di Ittiovirologia, Istituto Zooprofilattico Sperimentale delle Venezie, 35020 Padova, Italy; 3National Reference Centre for Fish, Mollusc and Crustacean Diseaseas, Istituto Zooprofilattico Sperimentale delle Venezie, 35020 Padova, Italy; 4Direzione Generale, Istituto Zooprofilattico Sperimentale delle Venezie, 35020 Padova, Italy

**Keywords:** nanoparticles, PCBs, spICP-MS, GC-MSMS, mussels

## Abstract

**Simple Summary:**

The interaction of nanomaterials with pollutants in the marine environment might alter bioavailability, as well as toxicity, of both nanomaterials and pollutants, representing a risk, not only for marine organisms, but also for consumers through the marine food chain. The aim of this research was to study whether titanium dioxide nanoparticles affect bioaccumulation and toxicity of pollutants, such as non-dioxin-like polychlorinated biphenyls in edible mussels harvested in a controlled contaminated environment. The results highlighted that titanium dioxide nanoparticles do not affect non-dioxin-like polychlorinated biphenyls’ accumulation in mussels, as their concentration was comparable with or without nanoparticles. Titanium dioxides nanoparticles in combination with non-dioxin-like polychlorinated biphenyls amplified the toxic effect on mussels compared to the exposure of non-dioxin-like polychlorinated biphenyls. Finally, mussels were submitted to a seven-day depuration process to evaluate the elimination of accumulated chemicals. Most titanium dioxide nanoparticles were eliminated after depuration, and their presence had a synergistic effect on non-dioxin-like polychlorinated biphenyls, which were eliminated in a greater quantity compared to mussels exposed only to non-dioxin-like polychlorinated biphenyls. In any case, consumers might be exposed to TiO_2_NPs and ndl-PCBs (both concurrently and separately) if edible mussels, harvested in a contaminated environment, are consumed without a proper depuration process.

**Abstract:**

The interaction of nanomaterials with pollutants in the marine environment might alter bioavailability, as well as toxicity, of both nanomaterials and pollutants, representing a risk, not only for marine organisms, but also for consumers through the marine food chain.The aim of this study was to evaluate the effect of titanium dioxide nanoparticles (TiO_2_NPs) in terms of bioaccumulation and toxicity on Mediterranean mussels (*Mytilus galloprovincialis*) exposed to six-indicator non-dioxin-like polychlorinated biphenyls (ndl-PCBs). Mussels were exposed to ndl-PCBs (20 µg/mL) (groups 3–4) or to a combination of ndl-PCBs (20 µg/mL) and TiO_2_NPs (100 µg/mL) (groups 5–6) for four consecutive days. TiO_2_NPs was detected in groups 5–6 (3247 ± 567 and 1620 ± 223 µg/kg respectively), but their presence did not affect ndl-PCBs bioaccumulation in mussels. In fact, in groups 3–4, the concentration of ndl-PCBs (ranging from 3818.4 ± 166.0–10,176 ± 664.3 µg/kg and 2712.7 ± 36.1–9498.0 ± 794.1 µg/kg respectively) was not statistically different from that of groups 5–6 (3048.6 ± 24.0–14,635.9 ± 1029.3 and 5726.0 ± 571.0–9931.2 ± 700.3 µg/kg respectively). Histological analyses showed alterations to the structure of the gill tissue with respect to the control groups, with more severe and diffuse dilatation of the central hemolymphatic vessels of the gill lamellae in groups 5–6 (treated with TiO_2_NPs and ndl-PCBs concurrently) compared to groups 3–4 (ndl-PCBs only). Finally, in mussels submitted to a seven-day depuration process, most TiO_2_NPs were eliminated, and NPs had a synergistic effect on ndl-PCBs elimination; as a matter of fact, in groups 5–6, the percentage of concentration was statically inferior to the one observed in groups 3–4. In any case, consumers might be exposed to TiO_2_NPs and ndl-PCBs (both concurrently and separately) if edible mussels, harvested in a contaminated environment, are consumed without a proper depuration process.

## 1. Introduction

Nanotechnology involves the manipulation of matter at the atomic or molecular scale, generally at the range below 100 nm. This gives different novel properties and functions to materials, which are referred to as nanomaterials (NMs) [1,2].

In particular, thanks to their extremely small size, NMs exhibit unique physicochemical properties, such as a high specific surface area and increased reactivity, which accounts for their widespread use in different industrial and biomedical applications, but also for new and unpredictable toxicological properties [1,2,3]. Different studies, conducted over the past decade, showed that the toxicity of most of the NMs studied is mainly related to reactive oxygen species (ROS) induced oxidative stress [4,5,6]. NMs can also exhibit toxicity by reacting with biological macromolecules or by releasing toxic components, such as metal ions [6].

The disposal of NMs leads to emissions to the environment, thereby increasing the potential for exposure of living organisms, including humans [7]. In particular, aquatic ecosystems, such as the marine environment, might be potential sinks for NMs, where the interaction with a variety of persistent organic pollutants (POPs) in the environment has recently aroused further concern [6,8,9].

Once in the aquatic system, NMs might be uptaken by aquatic organisms, and their uptake might be dependent on the interactions with other environmental pollutants. In particular, hydrophobicity of hydrophobic POPs allows them to bind with nanoparticles (NPs), and aquatic organisms might thus ingest organic chemicals attached on NPs [10,11,12] with alteration of bioavailability, as well as toxicity of both NPs and pollutants [6]. Bioaccumulated NPs and/or contaminants may transfer through food chains and cause ecological effects [6].

Discordant data are available regarding the combined effects of NPs and other contaminants in aquatic organisms, indicating that co-exposure to NPs may either amplify or alleviate the toxic effects of other compounds [6,8].

Moreover, the available results indicate that the interaction between NPs and contaminants depends, in addition to particles and contaminants properties, also on the different composition of water media (freshwater vs seawater). Most available data refer to freshwater, whereas few data are available about seawater [8]. Thus, further studies are needed to better study this complicated phenomenon.

Among NMs, titanium dioxide nanoparticles (TiO_2_NPs) are the most widely used, with many applications in different products, such as medicines, personal care products (e.g., sunscreens, toothpastes, cosmetics, soaps), plastics, paints, papers, sporting goods, self-cleaning surface coatings, solar cells, disinfectants, as well as in the environmental decontamination of air, soil, and water [13,14,15,16]. 

There is evidence that urban runoff waters and superficial waters are contaminated with TiO_2_NPs [7,17]. Furthermore, TiO_2_NPs are expected to be released in huge amounts in urban and industrial sewage and are estimated to reach a concentration of µg L^−1^ [8] in the aquatic environment.

Several studies suggest that TiO_2_NPs might interact with different contaminants, such as POPs, acting as carriers, affecting bioaccumulation and toxicity in several marine organisms [6,8,10,11,12,18].

For example, a significant increase in the toxicity of 2,3,7,8-tetrachlorodibenzo-p-dioxin (2,3,7,8-TCDD) has been observed in Mediterranean bivalve (*Mytilus galloprovincialis*) in the presence of TiO_2_NPs [19]. TiO_2_NPs also seem to alter the distribution, bioavailability, and toxicity of benzo(a)pyrene in the blue mussel *(Mytilus edulis)* [20]. Tian et al. [21] reported that TiO_2_NPs act as carriers and facilitate the bioaccumulation of phenanthrene in marine bivalves, i.e., ark shell (*Scapharca subcrenata*), and they enhance the uptake of polybrominated diphenyl ethers (PBDEs) by the marine bivalve (*Scapharca subcrenata*) [12].

Among POPs, polychlorinated biphenyls (PCBs) represent a class of 219 congeners, which were intentionally synthesized and massively used as dielectric and coolant fluids in the past [18,22]. Due to their environmental persistency and high toxicity, in 1979, their production was banned by the US Congress and, in 2001, they were also banned by the Stockholm Convention of POPs [23]. Although these preventive measures have led to a steady decrease in PCBs in the environment, as well as the reduction of their bioaccumulation in the marine food, PCBs are still present in the environment [18,22]. Moreover, they can still be released from poorly maintained hazardous waste sites, from the illegal disposal of PCBs-containing products, or leaks and releases from old electrical transformers containing PCBs [24]. The burning of some wastes in municipal and industrial incinerators and leakage from paint and sealants in older buildings represent a source of PCBs as well [18,22,24].

The peculiar chemical structure of PCBs affects their toxicological activity, so they can be divided in two main groups: non- or mono-ortho chloro substitutions are defined dioxin-like-PCBs (dl-PCBs), and the remaining congeners are defined as non-dioxin-like PCBs (ndl-PCBs) [18,22]. The Dl-PCB group includes 12 congeners, which exert their toxicity through the binding of aryl hydrocarbon receptor (AhR), such as polychlorodibenzodioxins (PCDD) and polychlorodibenzofurans (PCDFs). Ndl-PCB congeners have different toxicological pathways, which do not involve AhR and include neurological, neuroendocrine, endocrine, immunological, and carcinogenic effects [18,22].

The main source of human exposure to PCBs is via food consumption, with the exception of specific cases of accidental or occupational exposure. Different studies have proven that fish and fish-derived products are those with the highest PCBs contamination, with the ndl-PCBs being the most predominant congeners [18,22].

The likely interaction with NPs might modify the bioavailability and the bioaccumulation of these pollutants in the marine environment and open new exposure scenarios for marine species, but also for consumers exposed to contaminants through the aquatic food web.

The aim of this project was to conduct in vivo studies by exposing Mediterranean mussels (*Mytilus galloprovincialis*) to TiO_2_ NPs and to six indicator ndl-PCBs (PCB 28, 52, 101, 138, 153 and 180) in a controlled artificial marine environment, so as to obtain important clues to understand how their co-presence may affect their behavior and interaction in the aquatic environment in terms of toxicity in edible marine animals, but also in terms of bioaccumulation, which affects consumers’ exposure to these contaminants.

Single particle inductively coupled plasma mass spectrometry (spICP-MS) and gaschromatography coupled to mass spectrometry (GC-MSMS) were used as analytical tools to detect the residual presence of TiO_2_NPS and ndl-PCBs in mussels.

## 2. Materials and Methods

### 2.1. Standards, Chemicals, and Reagents

Powdered TiO_2_NPs (rutile) (diameter < 100 nm) were supplied by Sigma Aldrich (Saint Luis, MO, USA). A suspension of 60 nm gold nanoparticles and a mass concentration of 50 mg/L (AuNPs, RM8013) was purchased from NIST (Boulder, CO, USA) and used to determine the transport efficiency of the ICP-MS instrument [25,26]. A certified ionic titanium (Ti) standard (1000 mg/L) was obtained from ULTRA Scientific (North Kingstown, RI, USA) to prepare Ti calibration standards for spICP-MS analyses in the range of 0.15 to 2.5 μg/L.

Powders of certified analytical standard of 2,4,4′-Trichlorobiphenyl (PCB-28, purity 99.8 ± 0.5%), 2,2′,3,5,5′-Tetrachlorobiphenyl (PCB-52, purity 99.8 ± 0.5%), 2,2′,3,4,4,5′-Pentachlorobiphenyl (PCB-101, purity 99.9 ± 0.5%), 2,2′,3,4,4′,5′-Hexachlorobiphenyl (PCB-138, purity 99.9 ± 0.5%), 2,2′,4,4′,5,5′-Hexachlorobiphenyl (PCB-153, purity 98.3 ± 0.5%), and 2,2′,3,4,4′,5,5′-heptachlorobyphenyl (PCB-180, purity 99.1 ± 0.5) were purchased from Chiron AS (Trondheim, Norway).

For each ndl-PCB congener, a solution at 1 mg/mL in acetone was prepared by weighing 10 mg of powder in a 10 mL volumetric flask. 

A solution mixture of all the 6 ndl-PCBs congeners (concentration 0.05 mg/mL each) in water was prepared daily from the 1 mg/mL single solutions and used for in vivo treatment.

Certified analytical mix standard containing 6 ndl-PCBs congener (concentration 10 µg/mL each) in cyclohexane was supplied by Wellington Laboratories (Guelph, ON, Canada).

The corresponding labelled standards PCB 28-^13^C (2,4,4′–Trichloro [^13^C_12_] biphenyl), PCB 52-^13^C (2,2′,5,5′-Tetrachloro [^13^C_12_] biphenyl), (2,2′,4,5,5′-Pentachloro [^13^C_12_] biphenyl), PCB 138-^13^C (2,2′,3,4,4′,5-Hexachloro [^13^C_12_] biphenyl), PCB 153-^13^C (2,2′,4,4′,5,5′-Hexachloro [^13^C_12_] biphenyl, and PCB 180-^13^C (2,2′,3,4,4′,5,5′-Heptachloro [^13^C_12_] biphenyl)) single solutions (concentration 50 μg/mL in nonane e toluene (10%) each) were supplied by Wellington Laboratories (Guelph, ON, Canada).

Two intermediate mix solutions of all ndl-PCBs at 1 µg/mL and 100 ng/mL were prepared by mixing the appropriate amount of the corresponding solutions and diluting with HPLC-grade ACN.

An intermediate mix solution of the six labeled compounds, each at a concentration of 1 µg/mL, was prepared by appropriate dilution with HPLC-grade ACN of the pristine 50 μg/mL solution (Wellington Laboratories). An intermediate mix solution of all the labeled ndl-PCBs at 100 ng/mL was prepared by mixing the appropriate amount of the corresponding solutions and diluting them with HPLC-grade ACN.

Intermediate mixed solutions of ndl-PCBs and the corresponding labelled compound were used for GC-MSMS analysis.

Acetonitrile (ACN) (Ultra LC-MS grade) and acetone were purchased from VWR Chemicals (Radnor, PA, USA).

Proteinase K from Tritirachium album (lyophilized powder, BioUltra, ≥30 units/mg protein), sodium hexametaphosphate, Triton X-100, and Tris buffer were obtained from Sigma Aldrich (Saint Luis, MO, USA). A Milli-Q-Plus ultrapure water system from Millipore™ (Bedford, MA, USA) was used to produce Milli-Q-Water (MQW) for the preparation of samples and standards.

A Milli-Q-Plus ultrapure water system from Millipore™ (Bedford, MA, USA) was used to produce Milli-Q-Water (MQW) for the preparation of samples, standards, and other solutions for analysis. QuEChERS extraction salts (4 g Na_2_SO_4_, 1 g NaCl, 1 g trisodium citrate dehydrate, and 0.5 g disodium hydrogen citrate sesquihydrate) and dispersive solid phase extraction (D-SPE) (150 mg of primary secondary amine (PSA), 150 mg C18 and 900 mg of MgSO_4_) were supplied by Restek (Bellefonte, PA, USA).

### 2.2. TiO_2_NPs Dispersion Protocol

The procedure for dispersion of TiO_2_NPs in 2% sodium hexametaphosphate in water has been described before [7]. A suspension of TiO_2_NPs with a theoretical nominal concentration of 1000 mg/L was prepared daily, characterized in terms of total titanium (total-Ti) concentration and TiO_2_NP size distribution by spICPMS (Thermo Fisher Scientific Inc., Waltham, MA, USA) and used for in vivo treatment. The shape and size of the pristine powder have also been previously characterized in our laboratory [7] by transmission electron microscopy (TEM).

### 2.3. In Vivo Study Design

The in vivo experimentation lasted 16 days (five days of acclimatization, four days of treatment, and seven days of depuration) following a protocol previously developed by our group [7].

At least 700 mussels (*Mytilus galloprovincialis* (3–4 cm shell length)) were collected within a licensed mussel farming area in the southern part of Venice open sea (Italy), classified as A zone, according to EU Regulation 627/19 [27].

Twenty mussels were sampled upon arrival to be checked for pathogens (bacteria, parasites, viruses). All samples tested were free from the target pathogens.

All the remaining mussels were left for five days to acclimatize before starting the treatment. At the end of the 5th day, a sample of twenty mussels was submitted to chemical analysis, and no residue of TiO_2_NPs and ndl-PCMs were found.

Six experimental groups of mussels (100 individuals per group) were submitted daily to the following different treatments: two (control groups) (groups 1–2) were treated with 40 mL of a solution of sodium hexametaphosphate 2% (suspension stabilizer) and 16 mL of a aqueous solution of acetone 30%, two groups (groups 3–4 were) were treated with 16 mL of mix ndl-PCBs 0.05 mg/mL, and two groups (groups 5–6 were) were treated with 16 mL di mix ndl-PCBs 0.05 mg/mL and 40 mL of TiO_2_NPs suspension 100 mg/L.

The in vivo treatment lasted four days; freshly prepared treatment solutions were administered daily to the different mussel groups after the water change. At the end of the treatment, 45 mussels per tank group were collected for chemical analysis, and five mussels per group were collected for histological analysis.

The remaining 50 mussels of groups 1 and 2, groups 3 and 4, and groups 5 and 6 were merged to form three different groups, all of which were kept alive for a further seven days for a depuration study. During the depuration phase, water changes with clean artificial seawater were made daily. Finally, all the mussels were collected and submitted for chemical analysis.

### 2.4. Determination of TiO_2_NPs in Mussels by spICP-MS

The protocol for spICPMS analysis has been accurately described in [7]. Briefly, mussels (25 per group, without shells) were enzymatically digested and analysed by spICP-MS.

Quality controls (QC) included a procedural blank (QC-), a blank matrix (QCM-), and six positive QC samples (matrix matched samples) (QCM+) with TiO_2_NPs at 100 μg/kg.

Each QC sample was prepared, digested, and analyzed with each series of samples.

### 2.5. Determination of ndl-PCBs in Mussels by GC-MSMS

3.00 ± 0.05 g of homogenized whole mussel samples were spiked with 37.5 μL of IS (concentration 1 mg/L), mixed with 10 mL of MQW, and shaken on a vortex mixer (IKA Vibrax VXR (Staufen, Germany)) for 1 min. After the addition of 10 mL of ACN, samples were further shaken on an automatic stirrer (Genogrinder™, Spex^®^ Sample PREP, Stanmore, UK) at 25 Hz for 3 min. To induce phase separation and ndl-PCBs partitioning in the organic phase, QuEChERS extraction salts were added. The tubes were then closed and shaken again on an automatic stirrer at 25 Hz for 1 min and centrifuged for 5 min at 6000× *g* (Eppendorf Centrifuge 5810, Hamburg, Germany). All the extracts were submitted to D-SPE clean up. In detail, the extracts were transferred into a 15 mL plastic tube containing PSA, C18, and MgSO_4_, shaken on an automatic stirrer at 25 Hz for 3 min and centrifuged for 10 min at 6000× *g*. Finally, extracts were transferred to vials for GC-MSMS analysis.

QC included a procedural blank (QC-), a blank matrix (QCM-), and four positive QC samples (QCM+: two matrix matched samples, each spiked with PCBs at 6.25 μg/kg and two blank matrix matched samples, each spiked with PCBs at 1200 μg/kg).

Each QC sample was prepared and analyzed in each analytical batch to verify method performance and absence of contamination.

The analytical method was fully validated, and the following parameters were evaluated: specificity, linearity, recovery, and LOQ. The assessed LOQ for each PCB congener was 6.25 μg/kg. Recovery was always in the range between 80 and 120%. Detailed information about the validation of the method is presented in the Appendix A.

Analysis was performed on a Trace GC Ultra coupled with a TSQ triple quadrupole (QqQ) mass spectrometer instrument (Thermo Fisher Scientific Inc., Waltham, MA, USA) equipped with a split/splitless PTV injector and Triplus autosampler (Thermo Fisher Scientific Inc., Waltham, MA, USA).

The chromatographic separation was performed by a DB-5ms UI capillary column (30 m × 0.25 mm i.d., 0.25 μm film thickness; Agilent J & W, Santa Clara, CA, USA).The instrument control, data acquisition, and data analysis were performed using the Thermo Fisher Xcalibur software (Ver. 2.0). Helium was used as the carrier gas at a constant flow rate of 1.0 mL/min. Argon was used as the collision gas.

The PTV splitless injector conditions were: injection time of 0.02 min at 150 °C and pressure of 70 KPa, transfer time of 2 min with a final temperature of 300 °C (ramp temperature 14.5 °C/s) and pressure of 210 KPa, and a clean time of 10 min with a final temperature of 340 °C (ramp temperature 14.5 °C/s); the sample volume was 1 μL.

The column temperature was programmed as follows: initial temperature and time 170 °C and 1 min, respectively; first ramp of temperature at 8 °C/min and final temperature 280 °C; second ramp of temperature 15 °C/min and a final temperature of 300 °C kept constant for 5 min. The overall method time was 21 min.

The ion source and the transfer line temperature were set to 280 °C.

The operation conditions of the mass spectrometer were: electron impact ionization (70 eV) in selected reaction monitoring (SRM) mode; emission current 50 μA, electron multiplier voltage 1500 Volts; Q1 and Q3 peak width (FWHM) 0.70. Two SRM ion transitions (primary and secondary transitions of a precursor to product ions) for each PCB congener and one for each IS were determined via collision tests (Appendix A).

Matrix-matched calibration curves within the range of 0.75–100 μg/kg for the single congener (corresponding to 3–400 μg/kg in the sample) and internal standard correction were used for the quantification of the analytes. The introduction of the corresponding labelled internal standard for each analyte allowed suitable quantification, correcting analyte losses during the sample work-up, as well as matrix effects during measurement.

Whenever the concentration of the analysed sample was out the linearity range of the method, samples were properly diluted with the same matrix extract used to prepare matrix-matched calibration standards.

Xcalibur^TM^ software version 4.1, (Thermo Fisher Scientific, Bremen, Germany) was used to control the GC-MSMS system and to quantify analytes.

### 2.6. Histological Analysis of Mussels

Histological examination was performed on five specimens from each group at the end of the treatment period. Mussels were quickly cut open through sectioning the adductor muscle and transverse sections containing the mantle, gills, stomach, intestine, and digestive gland, which were fixed for 48 h in 4% buffered formaldehyde solution. Tissues were dehydrated by immersion in a graded series of ethanol and embedded in paraffin; 3 μm sections were obtained and stained with Harris’ haematoxylin and eosin-Y.

Slides were examined at 4×, 10×, 25×, 40× and 100× magnification with a Nikon H550L microscope to detect the presence of pathological alterations; the morphology of the digestive gland, as well as the mantle and gills and epithelium were compared with the different treatment groups. Digital images were obtained using an integrated Nikon DS Ri2 camera and NIS Elements BR software Version 5.30.03 (Nikon Europe BV, Amsterdam, The Netherlands).

### 2.7. Data Analysis

Statistical analysis was performed with Social Science Statistic (https://www.socscistatistics.com/, accessed on 14 November 2022). To determine statistically significant differences between the exposure groups, one way ANOVA followed by Tukey post hoc test was applied after testing the data for normality and homogeneity of variances. For datasets showing a non-normal distribution, Kruskal–Wallis one way analysis was applied. Significance values were set to *p* < 0.05. Graphs were prepared with Microsoft Excel 2013.

## 3. Results

### 3.1. Sp-ICPMS Characterization of TiO_2_NPs in Pristine Suspension Used in the In Vivo Study

The average, mode, and median diameters obtained by spICP-MS analysis (Figure 1) were 125, 90, and 130 nm, respectively, and the size distribution is shown in Figure 2.

### 3.2. Determination of TiO_2_NPs in Mussels by spICP-MS

Detailed results about TiO_2_NPs detection in mussels are reported in Table 1.

Figure 3 presents spICP-MS time scans of (a) a mussel sample treated by TiO_2_NPS and ndl-PCBs (group 5), (b) a mussel sample treated by ndl-PCBs only (group 3), and (c) a positive QC sample.

In Figure 4, TiO_2_NPs concentrations calculated by spICP-MS in mussels (group 5 and 6 merged) before and after the depuration process are presented.

### 3.3. GC-MSMS Determination of ndl-PCBs in Mussels

Detailed results about ndl-PCBs detection in mussels are reported in Table 2.

The concentration of ndl-PCBs in groups 3 and 4 (treated with ndl-PCBs only) is not statistically different (*p* < 0.05) from the one detected in groups 5 and 6 (treated with ndl-PCBs and TiO_2_NPs).

In Figure 5 and Figure 6, a comparison between ndl-PCBs concentration in mussels treated with ndl-PCBs only (groups 3 and 4 merged) and mussels treated with ndl-PCBs and TiO_2_NPs (group 4–5 merged), before and after the depuration process, are presented. In each figure, the percentage of ndl-PCBs detected after the depuration process is also reported.

Concerning groups 3 and 4, there is a statistical difference (*p* < 0.05) in terms of concentration only for PCB180 before and after the purification process.

Concerning groups 5 and 6 (TiO_2_NPs and ndl-PCBs treatment), there is a statistical difference before and after the purification process in terms of concentration (*p* < 0.05) for all congeners except for PCB-28.

### 3.4. Histological Analysis of Mussels

Histological examination showed alterations of the structure of the gill tissue in the treatment groups, in comparison with specimens in the control groups (Figure 7). In particular, dilatation of the central hemolymphatic spaces of the gill filaments was observed, which was moderately present in ndl-PCB-exposed mussels from groups 3 and 4 (50% of specimens showed multifocal dilatation, and 10% showed diffuse dilatation to most ctenidial filaments; Figure 7(B1,B2)) and of greater severity in ndl-PCBs + TiO_2_NP-exposed mussels from groups 5 and 6 (60% severe and multifocal dilatation; 30% severe and diffuse dilatation; Figure 7(C1,C2)). A high variability in the morphology of the digestive gland tubules was found among specimens housed in the same tank (flattening of the epithelium, high or low presence of vacuolization, dilatation of the tubule lumen) and even in specimens sampled from the control groups. The gonadal developmental stages of the tested mussels varied among specimens sampled from the same treatment group, and likewise the extent of reserve tissue in the mantle (adipogranular cells) was variable. Most specimens had gonads in an advanced stage of maturation.

## 4. Discussion

TiO_2_NPs used in the in vivo experiment were characterized by spICP-MS (Figure 1 and Figure 2). The average, mode, and median diameters of the TiO_2_NPs (125, 90, and 130 nm, respectively) were in line with those obtained in another study conducted in our laboratory [7], confirming the efficacy of the dispersion protocols used.

During the in vivo experimentation, mussels were checked several times daily by professional personnel to highlight any abnormal behavior (closed valves, damaged shells, filtration issues, etc.), and no mortalities were recorded during all the duration of the trial.

After in vivo experimentation, mussel samples were analyzed with spICP-MS to detect the presence of TiO_2_NPs (Table 1). No TiO_2_NPs were detected in control groups (groups 1 and 2), in groups treated with ndl-PCBs only (groups 3 and 4), or in negative QC-processed samples (QC-, QCM-). As expected, TiO_2_NPs were detected in all TiO_2_NPs-treated groups (groups 5 and 6) and in positive QC samples (QCM+, TiO_2_NPs 100 μg/kg).

The average size of the detected particles in groups 5 and 6 ranged from 123 to 132 nm, which is consistent with the size of the particles in the pristine TiO_2_NP suspensions used for the in vivo treatment and spiked into the QC samples (QCM+ TiO_2_NPs 100 μg/kg). The recovery of TiO_2_NPs in the positive QC samples ranged from 96 to 102%, while the determined particle sizes of 115 and 121 nm were in good agreement with the actual sizes of the pristine TiO_2_NPs. This result seems to confirm that TiO_2_NPs do not undergo any transformation during the bioaccumulation process in mussels [7].

Figure 3a–c, show the spICP-MS analysis of a 1000-fold diluted digested sample of group 5 (treated with TiO_2_NPs and ndl-PCBs), group 3 (treated with ndl-PCBs only), and positive QC samples (QCM+, TiO_2_NPs 100 μg/kg), respectively. Figure 3a,c present the typical spikes indicative of NPs’ presence. Moreover, the number of spikes present in the digested sample of group 5 is superior to the QC samples, as the NPs concentration is higher (Table 1).

As for ndl-PCBs, the GC-MSMS analysis revealed their presence in all the treated groups (3, 4, 5, 6) and in positive QC samples (QCM+ ndl-PCBs 6.25 µg/kg), whereas no ndl-PCBs were detected in the negative QC processed samples (QC-, QCM-).

Apparently, the presence of TiO_2_NPs does not seem to affect bioaccumulation of ndl-PCBs in mussels. In fact, ndl-PCBs concentration is not statistically different (*p* < 0.05) between groups 3–4 (treated with ndl-PCBs only) and groups 5–6 (treated with TiO_2_NPs and ndl-PCBs concurrently) for all the six congeners considered (Table 2). This result is in agreement with a study conducted on Mediterranean mussels (*Mytilus galloprovincialis*) at similar conditions with 2,3,7,8-tetrachlorodibenzo-p-dioxins [19] and to another study conducted on bivalves *Scapharca subcrenata* exposed to TiO_2_NPs and polybrominated diphenyl ethers (PBDEs) [12]. Interestingly, in both cases, the organic contaminants have chemical structure similar (with halogenated compound) to ndl-PCBs. Other few available studies on marine mussels, focusing on the concurrent exposure to TiO_2_NPs and other kinds of hydrophobic POPs, such as phenanthrene [21] and benzo(a)pyrene [20], provided contrasting results. In the first case, in fact, TiO_2_NPs enhanced phenanthrene uptake in ark shell (*Scapharca subcrenata*), and in the second study, benzo(a)pyrene concentration was reduced in the presence of TiO_2_NPs in blue mussel (*Mytilus edulis*). Different factors may play an important role in affecting the final results, and, in fact, in the two aforementioned studies, different kinds of mussels and POPs (e.g., without halogenated compound), compared to our study, were used.

On the other hand, histological analysis showed (Figure 7) a synergistic toxic effect due to the concurrent exposure to TiO_2_NPs and ndl-PCBs. In fact, an alteration in the gill filament architecture with more severe and diffuse dilatation of the central hemolymphatic vessels of the gill lamellae was observed in groups 5 and 6 (treated with TiO_2_NPs and ndl-PCBs concurrently) compared to groups 3 and 4 (ndl-PCBs only) and to the control groups.

Different studies showed a toxic interaction between NPs and co-contaminants [6], and modification of the normal gill structure after exposure to chemical stressors, such as heavy metals, microplastics, and pesticides, has been frequently described in the literature in mussels [28] mostly with the alteration or exfoliation of the ciliate epithelium and the dilatation of the lamellar lumen.

In the final part of the in vivo experiments, depuration of the TiO_2_NPs and ndl-PCBs by animals was investigated. Half of the mussels of merged group 3 and 4 (treated with ndl-PCBs only), as well as half of those of merged group 5 and 6 (treated with TiO_2_NPs and ndl-PCBs), were kept in clean artificial seawater for seven days.

Figure 4 shows how TiO_2_NPs concentrations decrease more than 70% after three days of depuration, in agreement with other studies reporting the ability of mussels to depurate metal NPs [7,29,30]. As for their capacity to eliminate ndl-PCBs, Figure 5 shows that, in groups 3–4 (treated with ndl-PCBs only), there is no statistical difference in terms of concentration before and after the depuration process for all congeners, except for PCB180, whose concentration showed a 53% decrease. This result confirms that hydrophobic POPs link in a strong manner to the mussel tissue and that the depuration process is very poor and takes a long time [31,32]. As a consequence, seven days might not be sufficient to perceive ndl-PCBs depuration for all the six congeners in mussels.

Finally, as showed in Figure 6, in groups 5–6 (treated with TiO_2_NPs and ndl-PCBs), there is a statistical difference (*p* < 0.05) between ndl-PCBs concentration before and after the depuration process for all the six congeners (except for PCB28) compared to group 3–4. For all the six congeners, a decrease in terms of concentration up to or more than 35% is observed.

These results suggest that, besides an increase in toxic effect, TiO_2_NPs seem also to enhance depuration from contaminants, and this might also explain why the presence of TiO_2_NPs does not seem to affect bioaccumulation of ndl-PCBs in Mediterranean mussels. This result was also reporteded in a study by Tian et al. [12], where, in bivalves *Scapharca subcrenata,* PBDEs’ overall bioaccumulation was not increased because TiO_2_NPs enhanced contaminant depuration.

## 5. Conclusions

The results of this study show that the concurrent exposure to TiO_2_NPs and ndl-PCBs represents a risk for marine organisms, such as Mediterranean mussels. In fact, NPs exhibit a synergistic toxic effect with ndl-PCBs.

In any case, taking into account the exposure to these contaminants due to consumption of mussels, the study shows that consumers might be exposed to TiO_2_NPs and ndl-PCBs (both concurrently and separately) if edible mussels, harvested in a contaminated environment, are consumed without a proper depuration process.

The current results cannot be extended to all kinds of TiO_2_NPs. In fact, not only the chemical forms, but also specific shapes and dimensions [33], play important roles in the biological activity of NPs.

## Figures and Tables

**Figure 1 animals-13-01208-f001:**
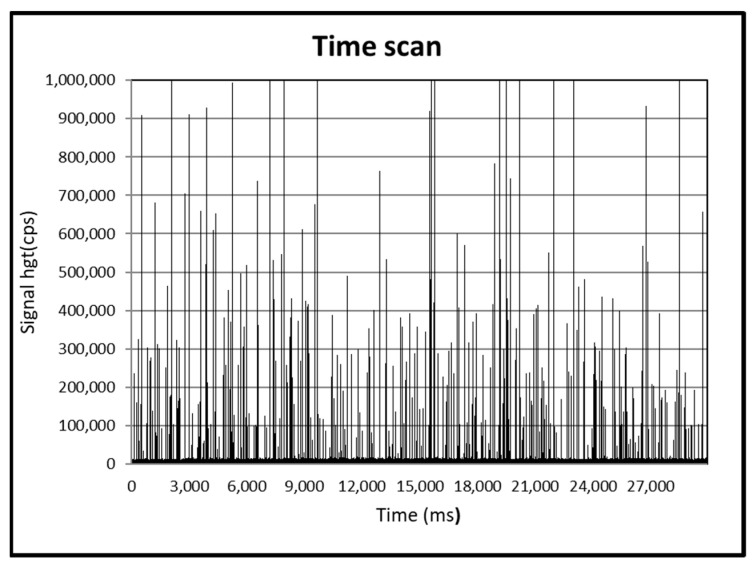
SpICP-MS time scans of pristine suspension of TiO_2_NPs.

**Figure 2 animals-13-01208-f002:**
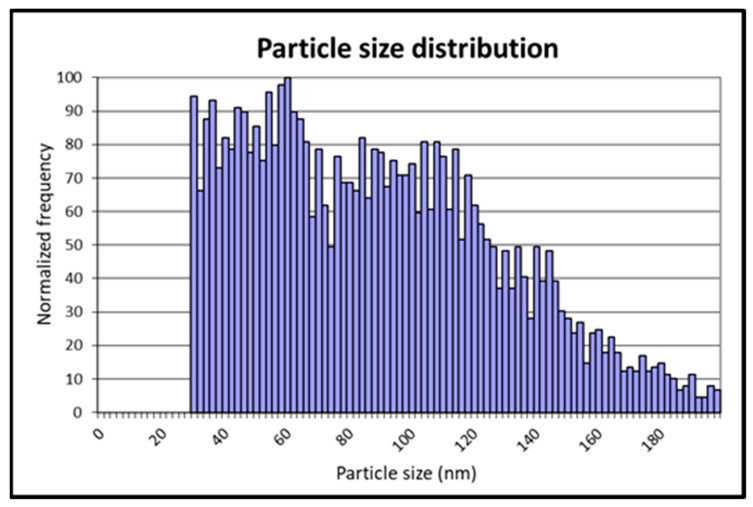
Size distribution of TiO_2_NPs pristine suspension obtained by sp-ICPMS.

**Figure 3 animals-13-01208-f003:**
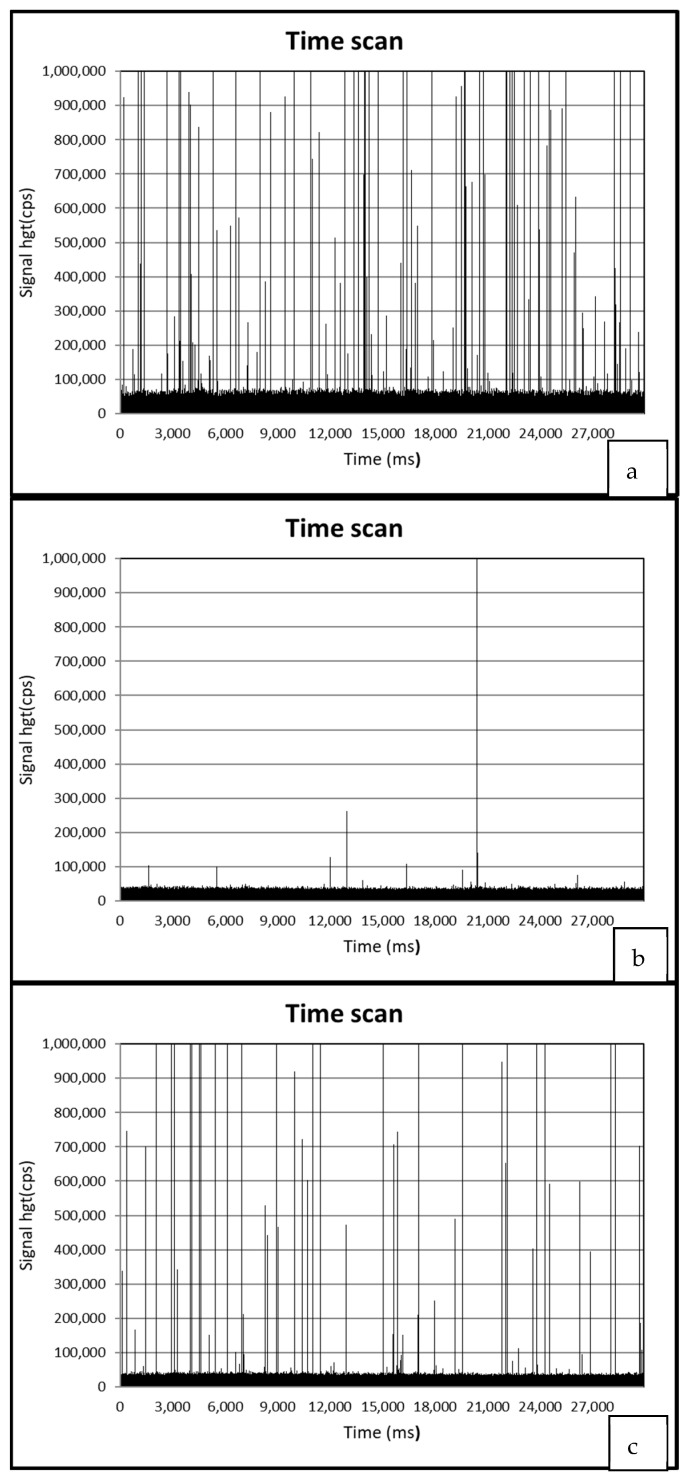
spICP-MS time scans of a mussel sample treated by TiO_2_NPs and ndl-PCBs (group 5) (**a**) and a mussel sample treated by ndl-PCBs only (group 3) (**b**) and a positive QC sample (**c**).

**Figure 4 animals-13-01208-f004:**
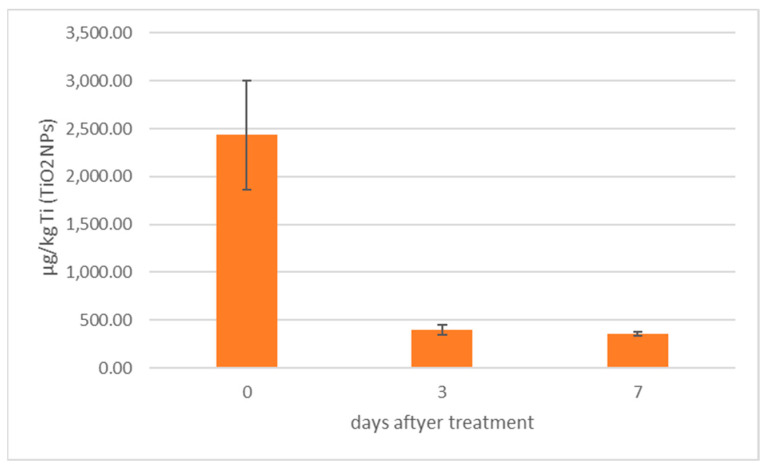
TiO_2_NPs concentration in mussels (group 5 and 6 merged) submitted to depuration process.

**Figure 5 animals-13-01208-f005:**
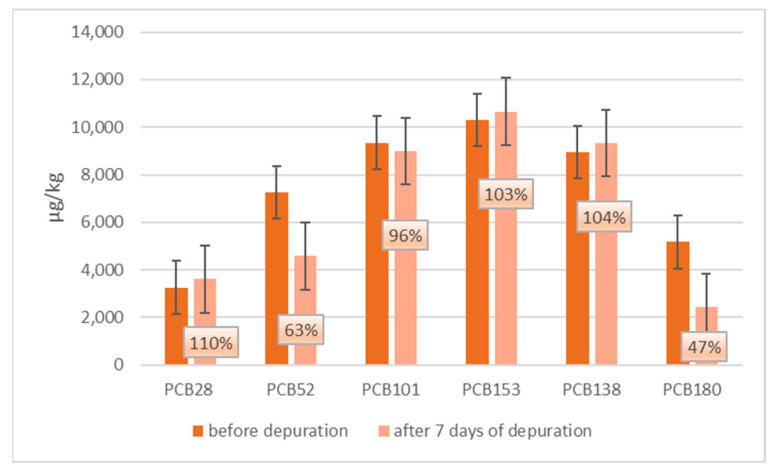
Ndl-PCBs concentration in mussels of merged group 3–4 (ndl-PCBs treatment only) before and after seven days of depuration process. The percentage of ndl-PCBs detected after the depuration process is also reported.

**Figure 6 animals-13-01208-f006:**
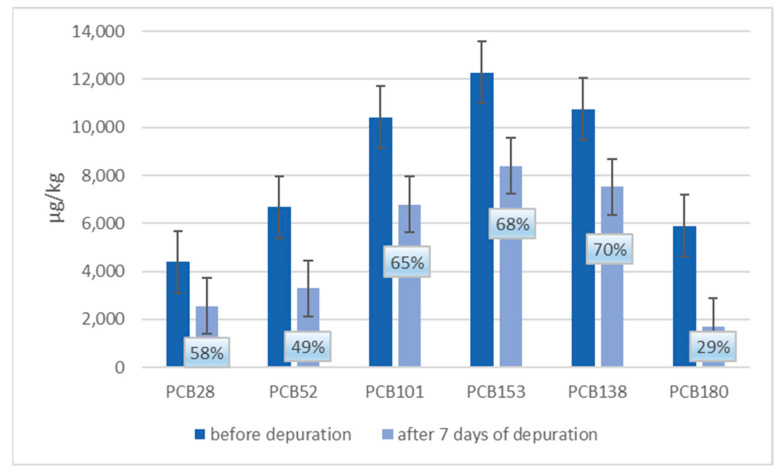
Ndl-PCBs concentration in mussels of merged group 5–6 (TiO_2_NPs and ndl-PCBs treatment) before and after seven days of depuration process. The percentage of ndl-PCBs detected after the depuration process is also reported.

**Figure 7 animals-13-01208-f007:**
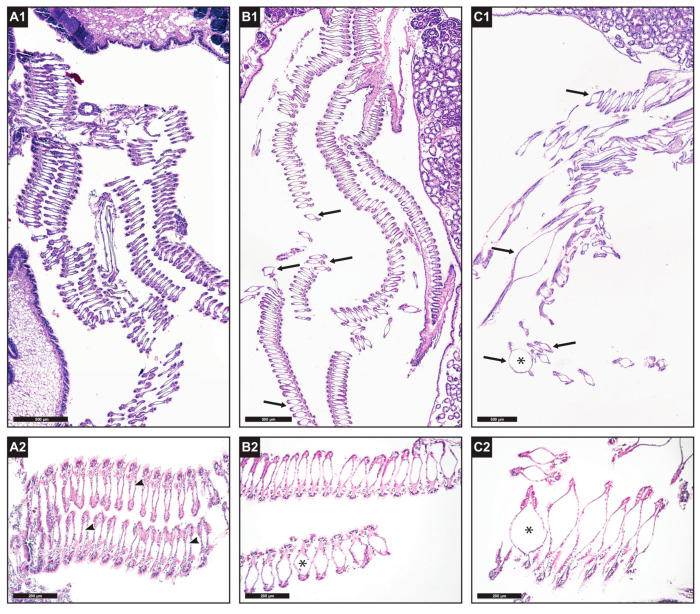
(**A1**): Gill ctenidial laminae from the control group, showing homogenous gill filament structure with hemolymphatic spaces of normal size (4×). (**A2**): detail of normal gill filaments from control (group 1–2) (10), hemolymphatic spaces appear normal (arrowheads). (**B1**): ctenidial laminae from the ndl-PCBs-treated group (4×), moderate dilatation of hemolymphatic spaces (arrows). (**B2**): detail of gill filaments from the ndl-PCBs-treated specimens (group 3–4) with moderate dilatation of the hemolymphatic spaces (*) (10×). (**C1**): ctenidial laminae of TiO_2_NPs and ndl-PCBs -treated group (4×), severe dilatation of the hemolymphatic spaces (arrows). (**C2**): detail of gill filaments of TiO_2_NPs and ndl-PCBs-treated specimen (group 5–6) with strongly dilated (*) hemolymphatic space (10×).

**Table 1 animals-13-01208-t001:** Mass-based concentrations and particle sizes of TiO_2_ NPs detected by spICP-MS in mussels after in vivo exposure to different treatments (with controls).

Sample	spICP-MS Analysis	
	Mass-Based Concentration of TiO_2_NPs in μg Ti/kg (N = 5) (Average ± SE) Ti^48^	Average Diameter of TiO_2_NPs in nm (N = 5) (Average ± SE)
Group 1 (control)	<LoQ = 50	<LoDsize = 40
Group 2 (control)	<LoQ = 50	<LoDsize = 40
Group 3 (ndl-PCBs 20 µg/mL)	<LoQ = 50	<LoDsize = 40
Group 4 (ndl-PCBs 20 µg/mL)	<LoQ = 50	<LoDsize = 40
Group 5 (ndl-PCBs 20 µg/mL + TiO_2_NPs 100 µg/mL)	3247 ± 567	132 ± 15
Group 6 (ndl-PCBs 20 µg/mL + TiO_2_NPs 100 µg/mL)	1620 ± 223	123 ± 20
QC-	<LOQ	na
QCM-	<LOQ	na
QCM+ (TiO_2_NPs 100 μg/kg)	101 ± 10	115 ± 10
QCM+ (TiO_2_NPs 100 μg/kg)	98 ± 9	121 ± 17

**Table 2 animals-13-01208-t002:** Concentrations of six indicators ndl-PCBs detected by GC-MSMS in mussels after in vivo exposure to different treatments (with controls).

Sample	GC-MSMS Analysis					
	PCB-28 µg/kg (N = 5) (Average ± SE)	PCB-52 µg/kg (N = 5) (Average ± SE)	PCB-101 µg/kg (N = 5) (Average ± SE)	PCB-153 µg/kg (N = 5) (Average ± SE)	PCB-138 µg/kg (N = 5) (Average ± SE)	PCB-180 µg/kg (N = 5) (Average ± SE)
Group 1 (control)	<LOQ = 6.25	<LoQ = 6.25	<LoQ = 6.25	<LoQ = 6.25	<LoQ = 6.25	<LoQ = 6.25
Group 2 (control)	<LoQ = 6.25	<LoQ = 6.25	<LoQ = 6.25	<LoQ = 6.25	<LoQ = 6.25	<LoQ = 6.25
Group 3 (ndl-PCBs 20 µg/mL)	3818.4 ± 166.0	8078.2 ± 293.5	10,176.3 ± 664.3	11,127.5 ± 36.0	9594.3 ± 308.0	5377.8 ± 698.1
Group 4 (ndl-PCBs 20 µg/mL)	2712.7 ± 36.1	6439.6 ± 175.2	8518.5 ± 119.2	9498.0 ± 794.1	8332.3 ± 10.6	4959.4± 60.9
Group 5 (ndl-PCBs 20 µg/mL + TiO_2_NPs 100 µg/mL)	3048.6 ± 24.0	7448.0 ± 350.3	11,560.5 ± 319.8	14,635.9 ± 1029.3	12,086.1 ± 403.4	6540.3 ± 282.3
Group 6 (ndl-PCBs 20 µg/mL + TiO_2_NPs 100 µg/mL)	5726.0 ± 571.0	5896.5 ± 54.7	9283.4 ± 920.3	9931.2 ± 700.3	9442.8 ± 110.9	5257.0 ± 67.1
QC-	<LOQ = 6.25	<LOQ = 6.25	<LOQ = 6.25	<LOQ = 6.25	<LOQ = 6.25	<LOQ = 6.25
QCM-	<LOQ = 6.25	<LOQ = 6.25	<LOQ = 6.25	<LOQ = 6.25	<LOQ = 6.25	<LOQ = 6.25
QCM+ (ndl-PCBs 6.25 µg/kg)	6.0 ± 1.0	6.4 ± 0.7	5.9 ± 0.5	5.8 ± 0.6	6.2 ± 0.7	6.6 ± 0.6
QCM+ (ndl-PCBs 6.25 µg/kg)	6.1 ± 0.9	6.3 ± 0.8	5.8 ± 0.5	6.0 ± 0.8	6.0 ± 0.8	6.5 ± 0.7
QCM+ (ndl-PCBs 1200 µg/kg)	10,773.1 ± 161.0	8966.6 ± 377.6	11,278.1 ± 782.4	11,551.9 ± 442.9	11,585.3 ± 736.0	10,770.1 ± 971.2
QCM+ (ndl-PCBs 1200 µg/kg)	11,004.5 ± 220.0	9725.3 ± 122.7	13,319.4 ± 340.5	11,738.7 ± 1111.1	12,047.3 ± 349.7	12,558.0 ± 511.6

## Data Availability

Data are contained within the article and Appendix A.

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
