# Peer review of "Effect of TiO2 Nanoparticle on Bioaccumulation of ndl-PCBs in Mediterranean Mussels (Mitilus galloprovincialis)"

_animals, 2023, doi:10.3390/ani13071208_

Round 1
Reviewer 1 Report
Upon reviewing your manuscript intitled: Effect of TiO2 nanoparticle on bioaccumulation of ndl-PCBs in Mediterranean mussels (Mitilus galloprovincialis). I find your work interesting, but I do not believe it can be published in its current form. Therefore, some revisions must be done so that it may be published in this journal. I have the following comments and recommendations:
- Check in the abstract, the values presented with commas must be informed with periods. Examples: ranging from 3818,4 ±166,0 – 10176 ±664,3 µg/kg and 2712,7 ± 36,1 – 9498,0 ± 794,1 µg/kg respectively and other.
- The introduction of relevant background and research progress was not comprehensive enough. Some work suggestions for your introduction; https://doi.org/10.1590/1980-5373-MR-2019-0463, https://doi.org/10.3390/coatings13010004, https://doi.org/10.1016/B978-0-323-99922-9.00004-0.
- Which of the famous TiO2 phases are present in the material used in this study? XRD analysis will be important in this study.
- Check de value: 3 ± 0.05 g: Error and Significant Algorithms
- The quality of the figures is poor and should be improved
- Check the values in table 2, replace the commas by points
- It is appreciable that the authors have studied: Effect of TiO2 nanoparticle on bioaccumulation of ndl-PCBs in Mediterranean mussels (Mitilus galloprovincialis), but the results presented are qualitative, some hypothetical and speculative. A stronger discussion should be done on this study before it is considered for publication.
Author Response
see the uploaded word file

Reviewer 2 Report
This is an interesting work that aims to understand how TiO2 NPs and ndl-PCBs interact and how their presence may affect their behavior and interaction in the aquatic environment. However, authors should clarify and describe more clearly aspects related to the contextualization and justification of the study, as well as some aspects related to the results obtained.
The abstract must contain the synthesis of the most relevant of each one of the parts of the manuscript. Information on the contextualization and justification of the work must be included.
Authors should emphasize and show even more clearly the relevance and importance of this work for Public Health.
Where did the mussels used come from? How did the authors ensure that the mussels were healthy and free of pollutants?
The sentence “No mortality was observed under different experimental conditions.” should not be included in the methodology since it´s observed results (line 195).
Figure 4 should be edited to remove the title as they have figure captions. the poor definition in Figure 3 does not allow for the correct interpretation of the graphs. figure 6 has no units of measure.
The images in Figure 7 are not of sufficient quality to allow identification of the alterations described. The magnification scale used in each case is not appreciated. Arrows pointing to the lesions can be sensed, but they are not clearly appreciated.
In section 3.4. “Histological analysis of mussels”, in addition to indicating the percentages, the authors should offer quantitative or, at least, semi-quantitative data on the main histopathology findings.
What evidence do the authors rely on to make the following statement?: (line 423): “The results of this study show that NPs might interact with other contaminants in the 423 marine environment. This might represent a risk for marine organisms, as NPs might have a synergistic toxic effect.”
Author Response
see uploaded word file

Round 2
Reviewer 1 Report
This version may be considered for publication